# Rapid and sensitive detection of gram-negative bacteria using surface-immobilized polymyxin B

**Hyun-Jin Kang**[1�], **Sang-Hoon Lee**[1�], **Han-Shin Kim**[1], **Yong Woo Jung**[2], **Hee-Deung Park**[1]*

1 School of Civil, Environmental and Architectural Engineering, Korea University, Seongbuk-Gu, Seoul, South Korea, 2 Department of Pharmacy, Korea University, Sejong, South Korea

☯ These authors contributed equally to this work.
* heedeung@korea.ac.kr

**Data Availability Statement:** All Supporting Informations are available from the figshare database (https://doi.org/10.6084/m9.figshare.23715531.v3).

## Abstract

Although detection of gram-negative bacteria (GNB) in body fluids is important for clinical purpose, traditional gram staining and other recently developed methods have inherent limitations in terms of accuracy, sensitivity, and convenience. To overcome the weakness, this study proposed a method detecting GNB based on specific binding of polymyxin B (PMB) to lipopolysaccharides (LPS) of GNB. Fluorescent microscopy demonstrated that surface immobilized PMB using a silane coupling agent was possible to detect fluorescent signal produced by a single *Escherichia coli* (a model GNB) cell. Furthermore, the signal was selective enough to differentiate between GNB and gram-positive bacteria. The proposed method could detect three cells per ml within one hour, indicating the method was very sensitive and the sensing was rapid. These results suggest that highly multifold PMB binding on each GNB cell occurred, as millions of LPS are present on cell wall of a GNB cell. Importantly, the principle used in this study was realized in a microfluidic chip for a sample containing *E. coli* cells suspended in porcine plasma, demonstrating its potential application to practical uses. In conclusion, the proposed method was accurate, sensitive, and convenient for detecting GNB, and could be applied clinically.

## Introduction

Gram staining is an old method for distinguishing bacterial cells into two large groups, gram-negative (GNB) and -positive bacteria (GPB), under microscopic observation [1]. The method is based on the degree of staining with crystal violet dye, which is affected by chemical and physical differences of cell walls of the two groups of bacterial cells [1, 2]. Cell walls of GNB consist of a thin peptidoglycan layer and outer membrane, while those of GPB contain a thick peptidoglycan layer [2]. Crystal violet binds the thick peptidoglycan layers of GPB more firmly compared with the thin peptidoglycan layers of GNB [3], which provides basis for color differentiation of the two groups of bacteria.

**Funding:** This work was supported by the Korea Ministry of Environment as Projects for Developing Eco-Innovation Technologies (GT-11-G-02-001-3) and by Basic Science Research Program through the National Research Foundation of Korea (NRF) funded by the Ministry of Education (2020R1A6A1A03045059). The funders had no role in study design, data collection and analysis, decision to publish, or preparation of the manuscript.

**Competing interests:** The authors have declared that no competing interests exist.

Rapid and accurate detection of bacterial cells and gram differentiation in body fluids in patients who are suspected of bacterial infection is crucial for appropriate antibiotic treatment [4, 5]. If a patient is diagnosed to be infected with GNB, antibiotics specific for GNB instead of GPB should be administrated. Furthermore, it is also important to specifically detect GNB in food and water [4, 6] as well as in body fluids [7], as dead GNB release heat-stable lipopolysaccharides (LPS) [8] as well as some GNB are pathogens (e.g., *Legionella pneumophila*, *Pseudomonas aeruginosa*, *Vibrio cholerae*). LPS consist of a repetitive glycan polymer (O-antigen), an oligosaccharide (core), and multiple fatty acids with phosphorylated glucosamine disaccharide (lipid A) [9]. LPS are components of outer membrane of GNB [9], and reported to be important in maintaining structural stability of a GNB cell and in protecting a GNB cell from chemical attack [10]. On the other hand, LPS as endotoxins endanger human health, as they can cause various immune response and fever in human [11].

GNB cells have been routinely detected using the gram staining method. However, some bacterial cells are not easily differentiable mostly because the method is highly dependent on operator's skill [12]. Furthermore, estimation of their cell numbers in a sample is challenging, especially for the samples with low cell numbers. To overcome the tedious gram staining method, several alternative methods have been developed. Selective plates (e.g., MacConkey agar and Eosin methylene blue agar) can cultivate GNB cells exclusively in a sample [13]. This culture-based method has an inherent weakness, as some GNB cells are slow to grow or even unculturable [14]. Quantitative real-time polymerase chain reaction can be applied to detect GNB cells. This technique is also limited to accurate detection, as the technique is dependent on primer (or probe) specificity [15]. Recently tip enhanced Raman spectroscopy was proposed for more accurate detection of GNB [5], but it requires an expensive analytical equipment. It is thus needed rapid, sensitive, and convenient methods for detecting and identifying GNB in clinical and environmental samples.

GNB contain huge numbers of LPS in their outer membranes. For instance, an *Escherichia coli* (a GNB) cell has 2 x $10^6$ LPS at its outer membrane layer [16]. If it is possible to specifically sense the LPS, GNB cells can be detected more sensitively. Polymyxin B (PMB) is a lipopolypeptide antibiotic produced by *Bacillus polymyxa* [17], and known to specifically bind to LPS [18]. Amine groups of polymyxin B can bind to phosphate groups of lipid A of LPS electrostatically, while nonpolar parts of polymyxin B can interact with acyl chains of lipid A of LPS hydrophobically [19].

Accordingly, the authors aimed to develop a sensor system for detecting GNB based on the interaction between LPS and PMB. Specificity, sensitivity, rapidity, and other features of the sensor system were evaluated. Finally, a microfluidic chip was fabricated for a convenient detection of GNB, and demonstrated its applicability to clinical purpose.

## Materials and methods

### Bacterial strains

Five species of GNB (*Escherichia coli* strain K12, *Pseudomonas aeruginosa* PA14, *Sphingomonas paucimobilis* ATCC51231, *Pseudomonas alcaliphila* AL15-21T (JCM 10630T), and *Enterobacter cloacae* ATCC13047) and four species of GPB (*Staphylococcus aureus* ATCC6538, *Bacillus amyloliquefaciens* strain FXH73, *Bacillus thuringiensis* ATCC10792, and *Corynebacterium xerosis* ATCC7711) were used in this study. All of the species were cultured overnight in Luria-Bertani broth (1% tryptone, 0.5% yeast extract, 1% NaCl solution, pH = 7.2) using a shaking incubator at 100 rpm. After dilution of the culture to 0.1 of optical density using sterilized deionized water, centrifugation was performed to pellet the bacterial cells at 10,000 g for 90 sec. The pellet was suspended in phosphate-buffered saline (PBS) (pH = 7.4), and

centrifugation was performed at the same condition. Centrifugation and suspension of the cells were performed twice more. The suspended cells were used for verification of the proposed method.

## Immobilization of PMB on a glass surface

A microscope glass slide (76 mm x 26 mm x 1 mm, Paul Marienfeld, Lauda-Königshofen, Germany) was initially dipped into piranha solution (3 parts $H_2SO_4$ and 1 part 30% $H_2O_2$ (v/v)) for 30 min to remove organic residues on the glass surface and to hydroxylate the surface. The cleaned glass slide was then dipped into 3-aminopropyltriethoxysilane (APTES) solution (TCI, Tokyo, Japan) (2% (v/v) in 95% ethanol) for 30 min for silanization. After washing away unbound APTES three times with deionized water, the slide was cured in an oven at 80°C for 30 min [20]. The slide was then dipped into glutaraldehyde solution (1% (w/v) in PBS) for 30 min for a cross linking between APTES and PMB. 500 µl of PMB (Sigma Aldrich, St. Louis, MO, USA) solution (0.1 ng/ml) was dropped on the center of the glass slide for reacting PMB with glutaraldehyde for 1 hr. Afterward, the glass slide washed with PBS (pH = 7.4) supplemented with 0.05% tween 20 (PBST) three times to remove unbound PMB. All of the reactions were conducted at room temperature.

## Detection of GNB

A sample containing bacterial cells (1 ml) was dropped on the PMB-immobilized slide for 30 min for reacting PMB molecules with GNB. Unbound bacterial cells were then washed away three times with PBST. The slide treated with bacterial cells was reacted with PMB molecules conjugated with fluorescein isothiocyanate (FITC-PMB) (1 µg/ml in H2O) for 30 min, and then the slide was washed three times with PBST. All of the reactions were conducted at room temperature. FITC was conjugated to PMB at the AnaSpec (Fremont, CA, USA) following the manufacturer's protocol.

## Quantification of FITC-PMB bound to GNB

The quantity of FITC-PMB bound to GNB was analyzed using the VICTOR x5 multimode plate reader (Perkin Elmer, Waltham, MA, USA) with a cartridge fitted to the glass slide. An F485 excitation filter (478–492 nm) and F535 emission filter (523–548 nm) were used for detecting fluorescence signals. A tungsten-halogen continuous wave lamp (75 w, spectral range 320–1,000 nm) was used to generate the laser. Excitation energy was set at 7,680 V and emission time was set at 0.2 sec and detection mode was bottom.

## Fluorescence microscopy

One hundred and fifty µL of 4',6-diamidino-2-phenylindole (DAPI) (Thermo Fisher Scientific, Waltham, MA, USA) solution (25 µg/ml) was applied on the immobilized *E. coli* cells for 15 min. After washing with PBST three times, the *E. coli* cells were stained with 150 µL of FITC-PMB solution (1 µg/ml) for 30 min. After washing with PBST three times, the *E. coli* cells on a glass slide were dried in a desiccator at room temperature for 1 hr. LSM700 confocal laser scanning microscope (CLSM) (Carl Zeiss, Jena, Germany) was used to visualize *E. coli* cells stained with DAPI and FITC-PMB, respectively. Fluorescence microscopic images were taken with a 40 X objective lens (C-Apochromat 40×/1.20 W Korr M27, Carl Zeiss) and with the filter sets for DAPI (excitation 350 nm, emission 470 nm) and FITC (excitation 490 nm, emission 525 nm), respectively, and processed using the Zen 2011 program (Carl Zeiss).

### Fluorescence-activated cell sorting

Fluorescence-activated cell sorting (FACS) was used to enumerate *E. coli* cells for comparing between colony forming unit (CFU) assay and the proposed method. Initially, washed *E. coli* cells with PBS (2.5 ml) were stained with FITC using the LYNX rapid fluorescein antibody conjugation kit (Bio-Rad, Hercules, CA, USA) following the manufacturer's procedure. The *E. coli* cells stained with FITC were counted using the FACSCalibur (BD Bioscience, San Jose, CA, USA) with a filter set (excitation 488 nm, emission 533 nm). *E. coli* cells were flowed through a FACS tube with 22 μm core size with a flow rate of 66 μL/min for 15 min. The data were processed using the CellQuest Pro software (BD Scientific).

### Microfluidic chip

A microfluidic chip (width of 25 mm, length of 75 mm, and thickness of 1.6 mm) with two channels was designed. Each channel has width of 5.8 mm, length of 55 mm, and height of 0.05 mm, which could hold 70 μl of fluid. The microfluidic chip was prepared with top and bottom plates, and fabricated by injecting polymethyl methacrylate (PMMA) to a mold [21] in a local company (NanoEnTek, Seoul, Korea). PMB was immobilized on the top plate in the same way as described above (Immobilization of PMB on a glass surface section) except for dropping 5 μl of PMB solution (0.1 ng/ml). Afterward, the top and bottom plates were combined using a customized press at 0.5 MPa, and by injecting 1.5 μl of acetone at six points around the channel wall [22].

For evaluation of the fabricated microfluidic chip, *E. coli* cells suspended in porcine plasma (Sigma Aldrich) (70 μl) was loaded into the injection hole of a channel using a micropipette. After reacting for 30 min, the sample was removed from the vent hole by capillarity action established by dry paper towel. Unbound GNB was washed away by loading PBST (70 μl) in the injection hole using a micropipette and by removing it from the vent hole using dray paper towel. Afterward, FITC-conjugated PMB solution (50 μl) was loaded, allowed to react for 30 min, and washed away unbound FITC-conjugated PMB in the same way as described above. Once all reactions were complete, the fluorescence intensity was measured using the VICTOR x5 multimode plate reader (Perkin Elmer, Waltham, MA, USA).

## Results and discussion

### Verification of the proposed method

Verification of the proposed method for detecting GNB was conducted two folds (Fig 1). First, feasibility of the proposed method was tested. *E. coli* cells (a model GNB) processed by the proposed method were stained using DAPI, a fluorophore that exhibits a blue fluorescence for all bacterial cells including *E. coli* cells upon binding with DNA [23]. CLSM was utilized to visualize the localization of the *E. coli* cells (blue) and the PMB conjugated with FITC (green), respectively. When the DAPI image (left in Fig 1(A)) was overlain with the FITC image (middle in Fig 1(A)), complete overlap of the two images (right in Fig 1(A)) was observed. This result suggests that all of the *E. coli* cells could be successfully detected by the proposed method. It is also noted that FITC signal was strong enough to identify a single *E. coli* cell. This is due to large numbers of LPS in the outer membrane of an *E. coli* cell. LPS are reported to cover ∼ 40% of cell surface of GNB [24]. The numbers are around two million in an *E. coli* cell [16]. Therefore, each individual *E. coli* cell was capable of highly multifold FITC-PMB binding.

Second, specificity of the proposed method was tested. Given that PMB cannot react with LPS-absent GPB, this method should exclusively detect GNB in a given sample. For

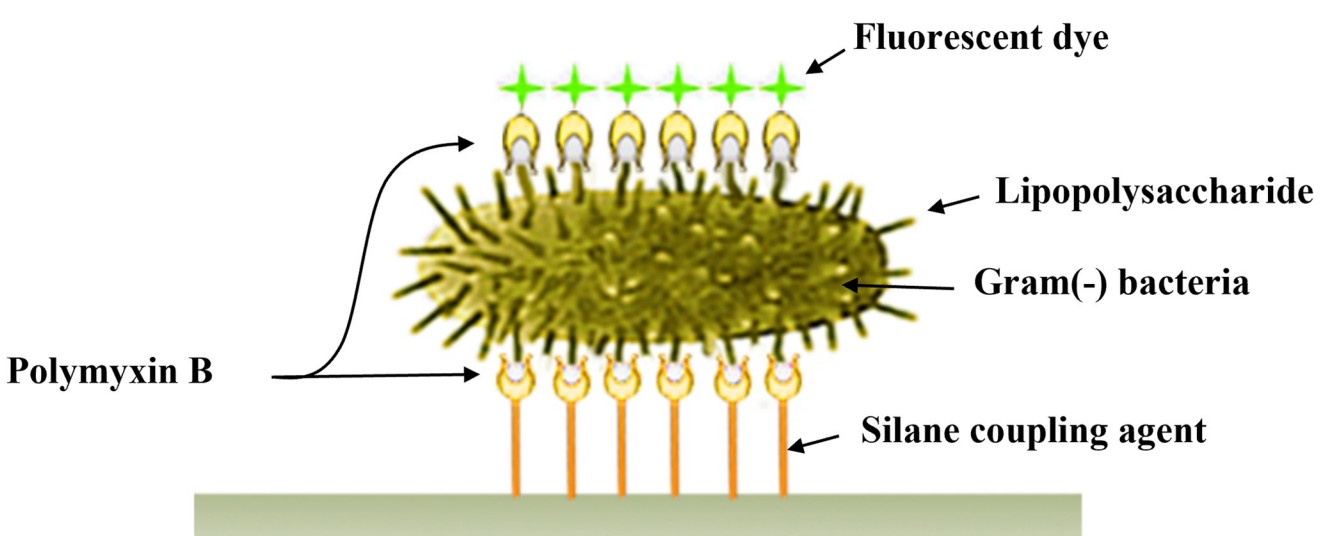

**Fig 1. Schematic of the principle for detecting GNB based on the binding of PMB to LPS.**

confirmation, experiments were conducted using five GNB species (*E. coli*, *P. aeruginosa*, *S. paucimobilis*, *P. alcaliphila*, and *E. cloacae*) and four GPB species (*S. aureus*, *B. amyloliquefaciens*, *B. thuringiensis*, and *C. xerosis*). As shown in Fig 2(B), FITC signal intensity of GNB (357–517 arbitrary units (AU)) was much higher than that of GPB (36–43 AU) (P < 0.001 in t-test). The fluorescence intensity of the GPB was similar to that of the negative control (40 AU) (P > 0.10 in t-test). These results demonstrate that the proposed method in this study was selective enough to detect GNB against GPB. However, the fluorescence intensities of GNB were different from each other. This might be caused by different amounts of LPS depending on species [25]. Furthermore, different amount of LPS among species appears to be associated with variability of their biomass. Watson et al. [26] showed an experimental data in which marine bacterial biomass varied a factor of three, and cell size affected the number of LPS (i.e., larger cells had more LPS). They also demonstrated that cell size can be varied in a same species depending on growth phase. When *E. coli* cells entered into a stationary growth phase, size of them decreased with decreasing LPS. Nevertheless, the ratio of bacterial carbon with LPS was kept to 6.35 irrespective of growth phases.

## Optimization of the reaction time for the proposed method

Optimized processing time for the proposed method was determined by changing reaction times. First, *E. coli* cells were bound on PMB-immobilized glass slides by increasing the binding time for 5 min over an hour, and FITC-conjugated PMB was then reacted with the *E. coli* cells bound to the glass slides for another hour. As shown in Fig 3, the FITC signal intensity gradually increased with the reaction time up to 20 min and afterwards the fluorescence intensity was saturated. This demonstrates that 20 min were sufficient for binding *E. coli* cells on the PMB-immobilized glass slides. Second, the minimum time for binding FITC-conjugated PMB to the *E. coli* cells which were already bound to glass slides for an hour was analyzed by increasing the reaction time for 5 min over an hour. Like the binding *E. coli* cells on the PMB-immobilized glass, the binding FITC-conjugated PMB to the *E. coli* cells bound to the glass

**(A)**

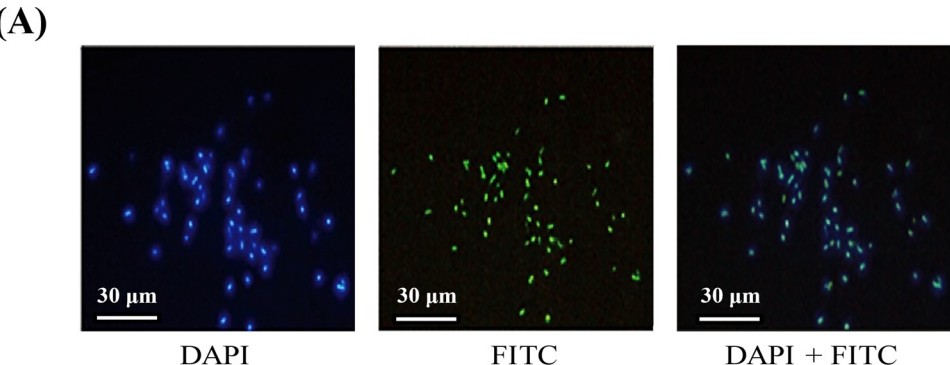

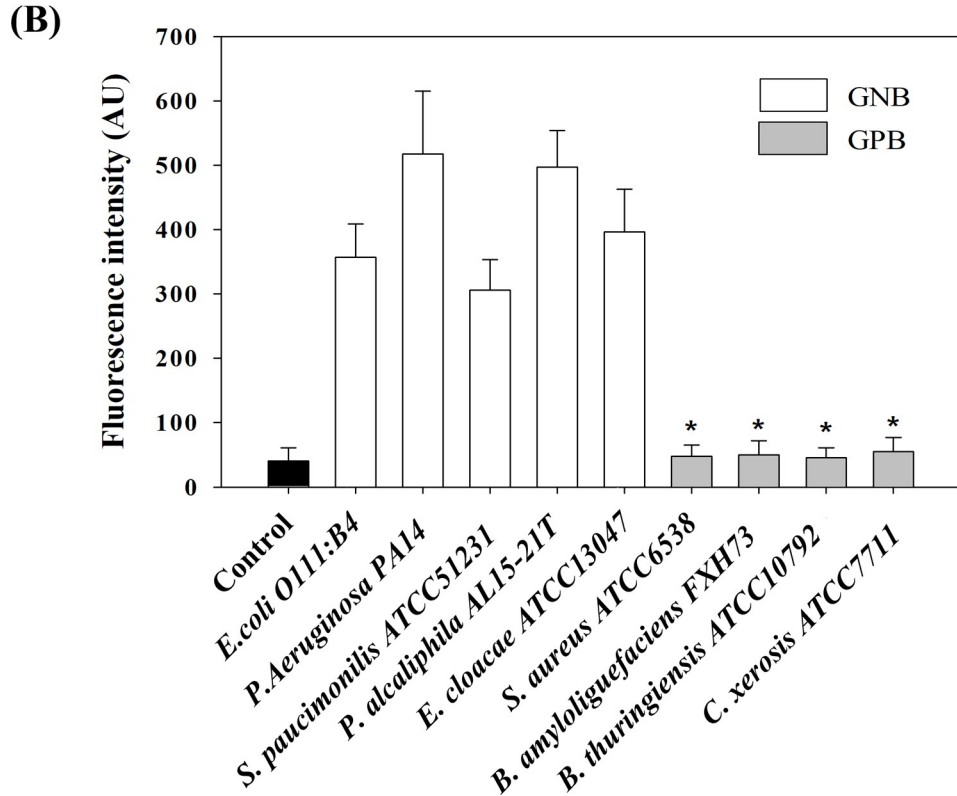

**Fig 2. Verification of the proposed method detecting GNB.** a) CLSM images of DAPI (left), FITC (middle), and overlay of the two images (right) for the *E. coli* cells processed by the proposed method and stained with DAPI. Scale bars indicate 30 μm. b) FITC signal intensity for various GPB and GNB processed by the proposed method. Control indicates a sample without bacterial cells. The error bars indicate the standard deviation of five measurements. *, P > 0.10 versus control.

slides was saturated after 20 min. This demonstrates that 20 min were sufficient for binding FITC-conjugated PMB to the *E. coli* cells bound to glass slides. Based on these results, we set reaction time as 30 min for binding GNB on PMB-immobilized glass slides and as 30 min for binding FITC-conjugated PMB on GNB bound on glass slides, respectively, for all experiments (i.e., one hour total). The time of analysis is comparable to that of real-time PCR and immune sensors [27, 28]. Interestingly, the trends of increasing fluorescent intensity for the two bindings were almost overlapped. This result suggests that the rate of binding *E. coli* cells on the

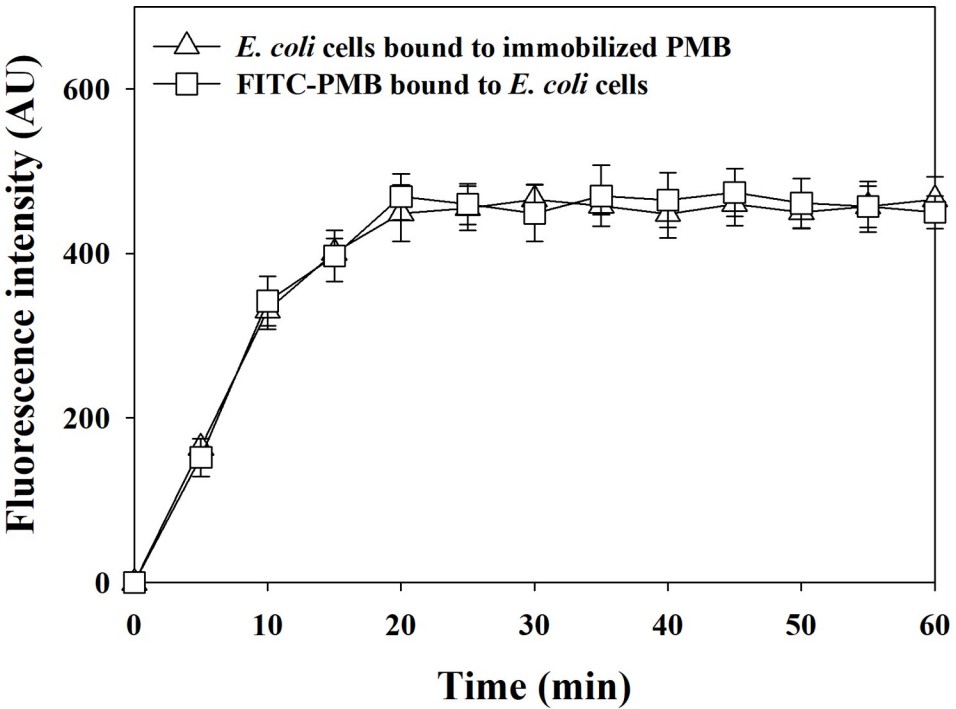

**Fig 3. Optimization of a reaction time of the proposed method.** FITC signal intensities were measured by increasing binding time for *E. coli* cells on the PMB-immobilized glass slides (opened square) and for FITC-conjugated PMB on *E. coli* cells bound to glass slides (opened triangle) for 5 min over an hour. The error bars indicate the standard deviation of five measurements.

PMB-immobilized glass and the rate of binding FITC-labelled PMB to the *E. coli* cells were kinetically similar.

## Sensitivity of the proposed method

Sensitivity of the proposed method was evaluated by comparing with a traditional method for quantification of microorganisms (CFU assay). pH and temperature may affect the sensitivity of the proposed PMB method, but experiments conducted for different pH (3, 7, 11) and temperature (4, 36, 60°C) for various GNB demonstrated insensitivity of the PMB method (S1 Fig). For the analysis, *E. coli* cells cultivated at 35°C for 12 hours was diluted using PBS (pH = 7.4) to generate 3–432 cells per ml, and their numbers were measured using the two methods. Linearity of the cell numbers (measured using FACS) with each method was estimated using regression analysis (Fig 4). Although the two methods showed fairly good linear relationships within the test range ($R^2 > 0.99$), they showed different sensitivity for low numbers of cells. As shown in the indented figures of Fig 4, the linearity decreased below $\sim 80$ cells for the CFU assay ($R^2 = 0.963$), but it was not deteriorated down to three cells for the proposed method ($R^2 = 0.997$). Detection limit of the proposed method is two to three orders higher than other methods for *E. coli* reviewed by Lazcka et al. [28]. They reported that the detection limits of real-time PCR techniques were $10^2$–$10^3$ cells/ml, except for the value of Fu and Kieft [29] (5 cells/ml), and those of fiber optic immunosensor, surface plasmon resonance biosensor, quartz crystal microbalance biosensor, conductimetric biosensors, and impedimetic immunosensors were $2.9 \times 10^3$, $10^2$, $10^3$, 79, and $10^4$ cells/ml, respectively. The higher sensitivity of the proposed method in this study was probably due to multifold FITC-PMB binding to *E. coli* cells as discussed earlier.

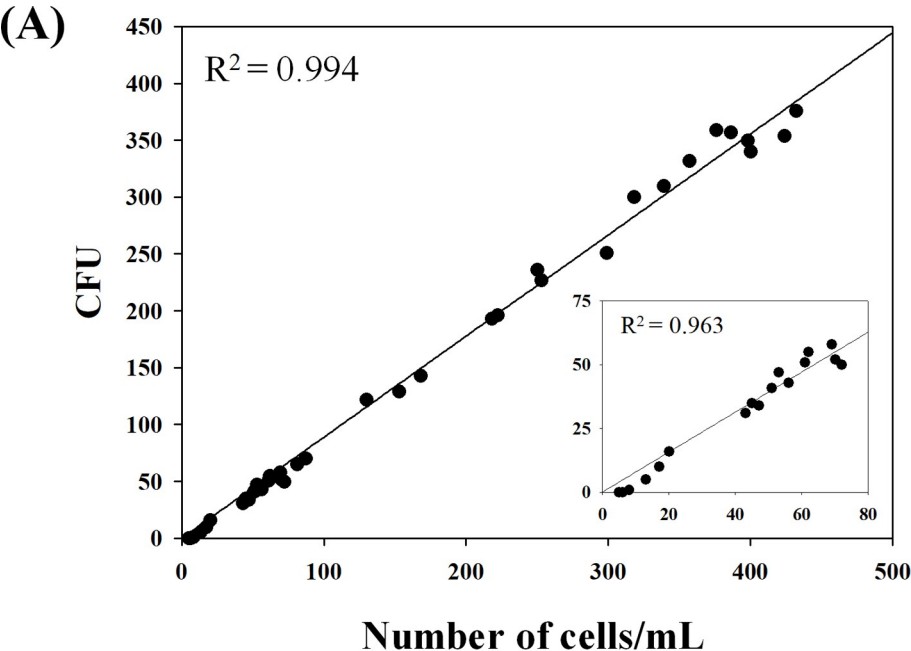

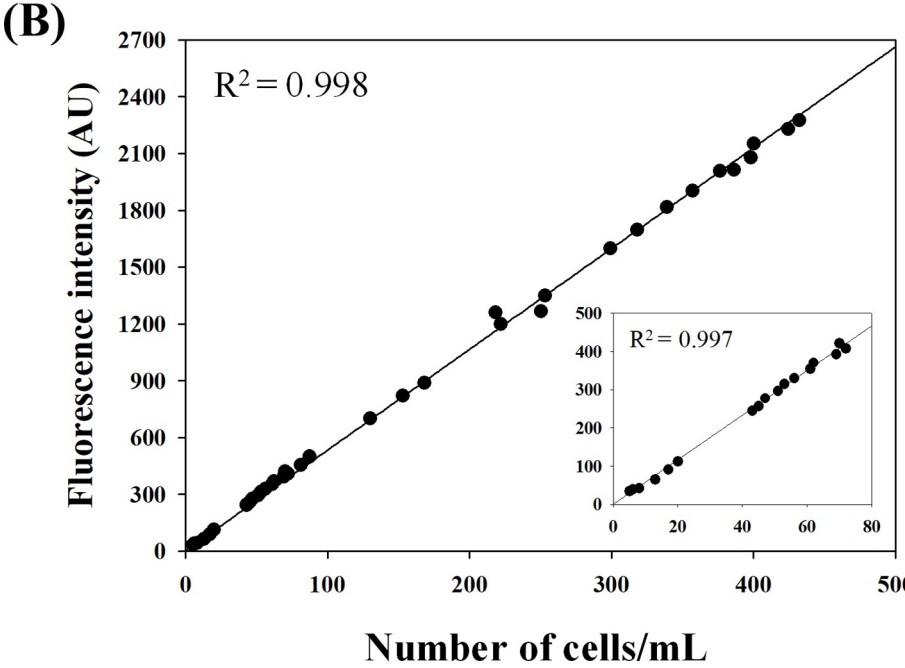

**Fig 4.** Comparison of sensitivity for detecting *E. coli* cells between CFU assay (a) and the proposed method (b). Number of *E. coli* cells in the samples was analyzed by FACS.

### Other features of the proposed method

In nature, most microorganisms exist in aggregated forms such as biofilms and flocs [30], rather than dispersed forms. An experiment was conducted to test whether the proposed method was robust enough in quantifying GNB with aggregated forms. To this purpose, the proposed method and the CFU assay were compared in terms of GNB detection for the

dispersed activated sludge cells and the activated sludge flocs. As shown in Fig 5, the CFU assay yielded 1,350 ± 178 CFU and 72 ± 16 CFU for the dispersed and aggregated activated sludge cells, respectively, amounting to a 19-fold difference. The large difference between the two forms of cells was due to the nature of the CFU assay which assumes that a single colony is propagated by a single cell. In the case of the aggregated activated sludge floc, this condition does not hold; thus, the measurement of the overall GNB cells was grossly underestimated. In case of the proposed method, the dispersed activated sludge cells resulted in 8,341 ± 731 AU; while, the aggregated activated sludge floc resulted in 7,187 ± 631 AU. Higher estimation of GNB for the dispersed cells than that for the aggregated flocs (although the difference between the two forms of cells was small) might be due to difficulty of diffusion of the FITC-conjugated PMB to the microorganisms deeply localized within the floc, resulting in lower fluorescence intensity of the aggregated cells compared to the dispersed cells. Nevertheless, when compared with the CFU assay, the PMB method showed greater sensitivity and lower underestimation.

As the proposed method detect LPS located at the outer membrane of GNB, it would be independent from the viability of GNB. In other words, the proposed method can detect dead GNB as well as live GNB. This possibility was tested using the 0.5% NaOCl-treated *E. coli* cells (i.e., dead *E. coli* cells) and the untreated *E. coli* cells (i.e., live *E. coli* cells). Fig 6 shows a comparison between the proposed method and the CFU assay for the two types of cells. The proposed method yielded values of 1,438 ± 198 AU and 1,583 ± 217 AU for the dead and live cells, respectively. The counting sensitivity of the dead cells represented by fluorescence intensity was slightly lower than that of live cells (91% of the live cells), presumably due to some dead *E. coli* cells ruptured and split LPS into the medium. While the plating method showed 9 ± 4 and 137 ± 17 CFU for the dead and live cells, respectively. It appears that dead *E. coli* cells could not grow and thus were not detectable in the CFU assay. However, the proposed method detected dead *E. coli* cells as the authors expected. LPS are major endotoxins that can elicit

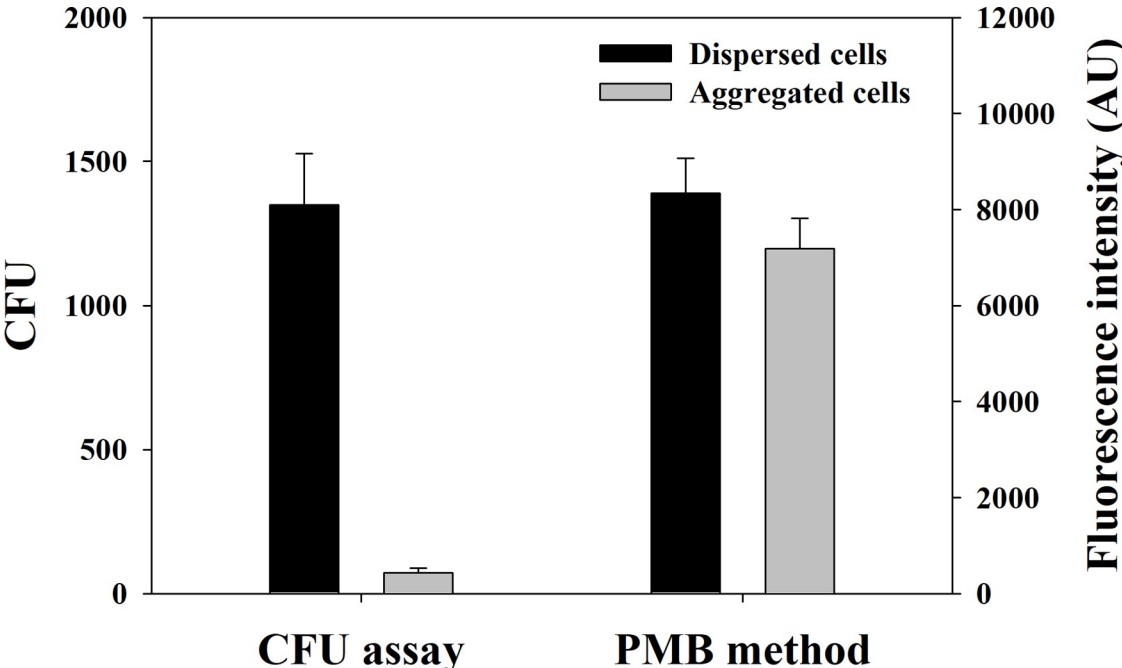

**Fig 5. Evaluation of cell aggregation for detecting GNB in the CFU assay and the proposed method.** Activated sludge floc was used a form of aggregated cells. The error bars indicate the standard deviation of five measurements.

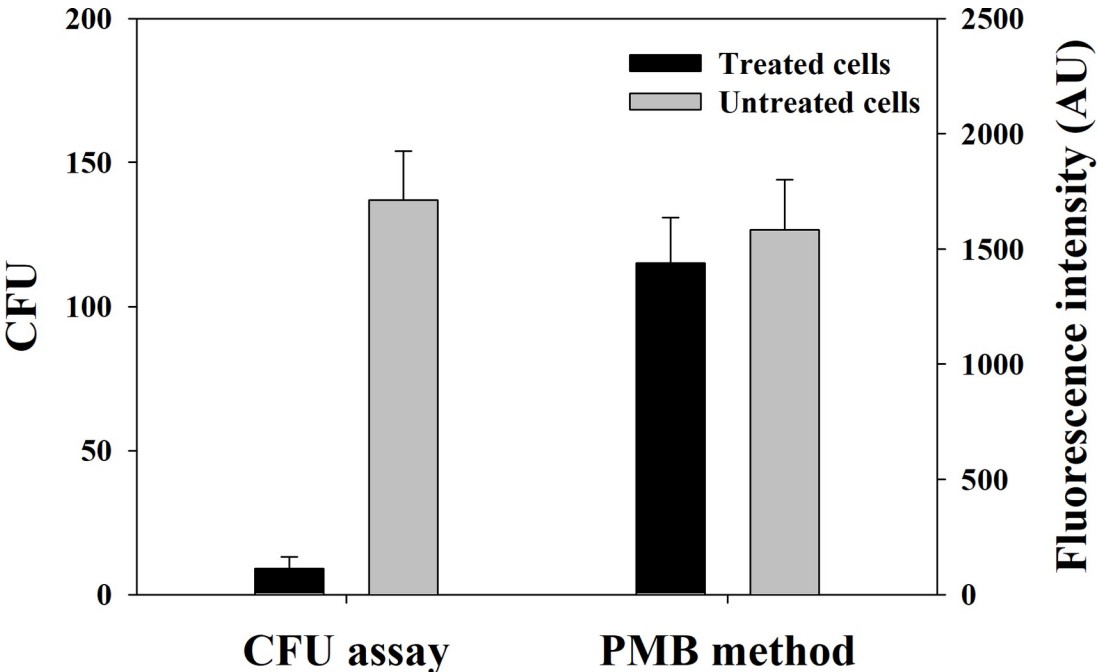

**Fig 6. Evaluation of viability of for detecting GNB in the CFU assay and the proposed method.** *E. coli* cells were killed using 0.5% sodium hypochlorite for a preparation of dead GNB cells. The error bars indicate the standard deviation of five measurements.

various immune responses via binding with toll-like receptors in human [31]. As such immune responses are initiated by LPS that are released by mostly dead GNB [8, 31], dead rather than live GNB pose more of a threat to human beings; thus, proper attention must be paid to dead GNB as well. In this respect, capability of dead GNB detection by the proposed method suggests it can be used for early detection of such risk elements.

## Detection of GNB in a microfluidic chip

A microfluidic chip adopting the principle proposed in this study was fabricated for convenient detection of GNB (Fig 7(A)). The microfluidic chip has two independent channels, and each channel (55 mm length, 5.8 mm width, 0.05 mm depth) had a dotting point where PMB was immobilized using APTES in the center of the channel. Detection of GNB using the microfluidic chip is achieved by following sequential steps: loading a sample solution, discarding the sample solution, washing the channel, loading a FITC-conjugated PMB solution, discarding the PMB solution, washing the channel, and measuring fluorescence intensity (detailed method is described in Microfluidic chip section).

In this study the authors would like to confirm applicability of the proposed method in detecting GNB that are present in body fluid using the fabricated microfluidic chip. *E. coli* cells diluted in porcine plasma were used to investigate the relationship between cell number (measured using FACS) and fluorescent intensity (Fig 7(B)). A regression analysis showed high linearity of the relationship ($R^2 = 0.988$) under the condition where the number of cells was small (4–91 cells per injection (70 μl)), although the linearity was slightly lower compared with the test conducted using a glass slide with *E. coli* cells diluted in PBS (Fig 4(B)). Probably, the components in porcine plasma affected the binding of GNB cells on the immobilized PMB and/or FITC conjugated PMB to GNB cells. Porcine plasma is reported to contain various proteins,

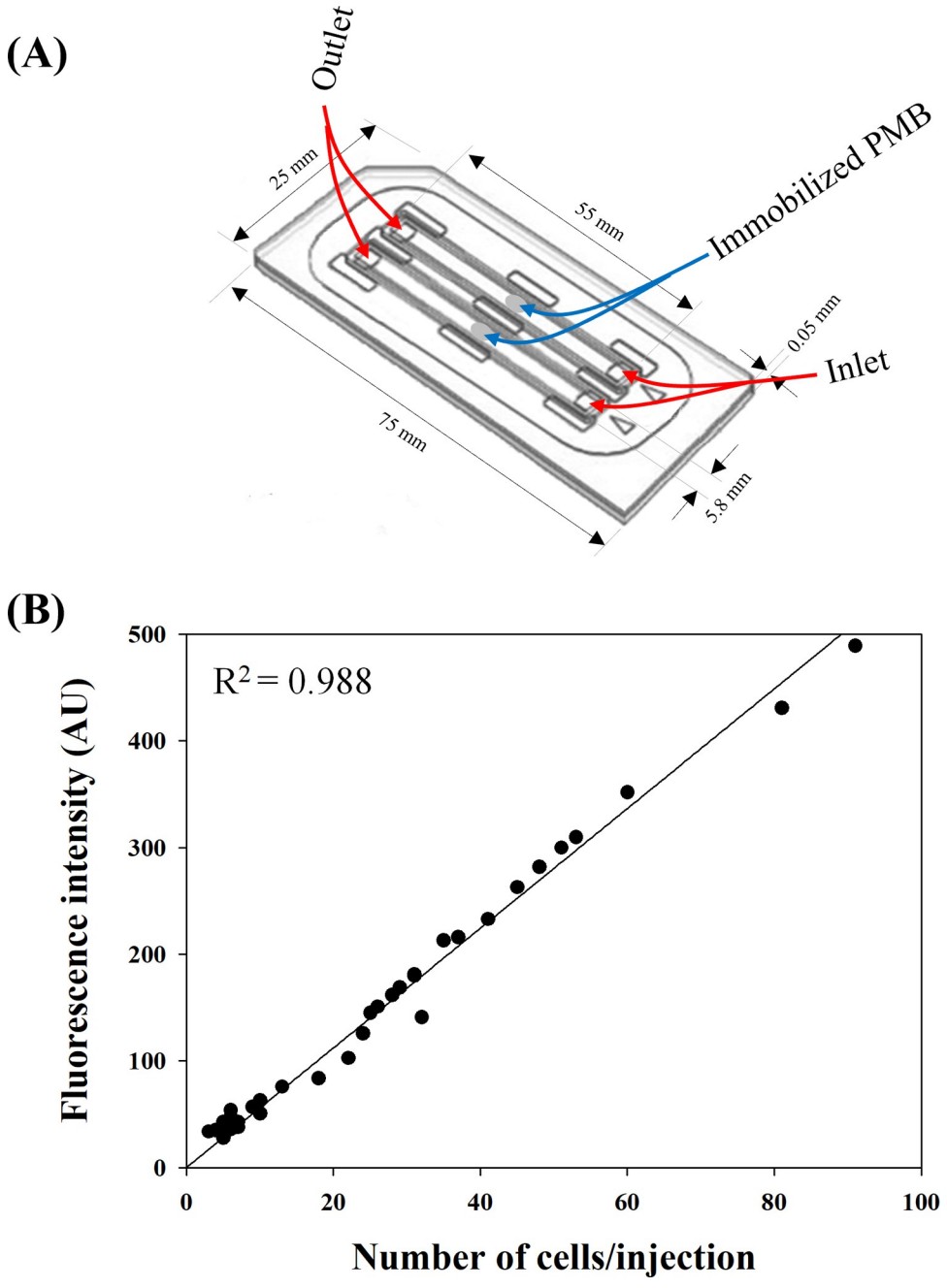

**Fig 7. GNB detection using a microfluidic chip.** a) Schematic of the microfluidic chip with two independent channels. Dimensions, injection hole, vent hole, and dotting points where PMB was immobilized are indicated. b) Relationship between the number of cells and FITC intensities analyzed in the microfluidic chip.

nutrients, and electrolytes [32]. In addition, the reaction condition for the detection of GNB in the microfluidic chamber might not be optimal. Nevertheless, the result (Fig 7(B)) suggests that the microfluidic chip was sensitive enough detect low numbers of GNB cells ($\sim$ 5 cells per injection) in body fluids.

## Conclusions

This work described a method detecting GNB based on specific binding of PMB to LPS in outer membrane of GNB. GNB were initially bound to surface immobilized PMB, and FITC-conjugated PMB was then bound to the bound GNB for the detection. The proposed method was selective to differentiate between GNB and GPB, and was sensitive enough to detect three *E. coli* cells per ml. The GNB detection could be achieved within an hour. Interestingly, the PMB-based method was not deteriorated in estimating cell numbers for aggregated cells, and could detect dead GNB cells. Furthermore, a microfluidic chip adopting the principle proposed in this study worked for low numbers of *E. coli* cells in porcine plasma. Taken together, this work strongly suggests that the PMB-based GNB detection method was robust enough to specifically detect low numbers of GNB cells in body fluids.

## Supporting information

**S1 Fig.** Sensitivity of the proposed PMB method under different (A) pH and (B) temperature conditions. The error bars indicate the standard deviation of five measurements.
(TIF)

## Author Contributions

**Conceptualization:** Hyun-Jin Kang, Han-Shin Kim.

**Data curation:** Sang-Hoon Lee, Yong Woo Jung.

**Formal analysis:** Sang-Hoon Lee, Han-Shin Kim, Yong Woo Jung.

**Funding acquisition:** Hee-Deung Park.

**Resources:** Hee-Deung Park.

**Supervision:** Hee-Deung Park.

**Visualization:** Hyun-Jin Kang.

**Writing – original draft:** Hyun-Jin Kang, Han-Shin Kim, Yong Woo Jung, Hee-Deung Park.

**Writing – review & editing:** Hyun-Jin Kang, Sang-Hoon Lee, Hee-Deung Park.

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
