## [Decision Letter · Decision Letter 0]

15 Jun 2023

PONE-D-23-10755Rapid and sensitive detection of gram-negative bacteria using surface-immobilized polymyxin BPLOS ONE

Dear Dr. Park,

Thank you for submitting your manuscript to PLOS ONE. After careful consideration, we feel that it has merit but does not fully meet PLOS ONE’s publication criteria as it currently stands. Therefore, we invite you to submit a revised version of the manuscript that addresses the points raised during the review process. In general the reviewers are positive relativelly to the quality of the presented work, but they did some suggestions that can improve the manuscript. Please try to follow all the suggestions that can contribute to that.One of the reviewers sent his comments in a separated word file, be sure that you received the comments of two reviewers.

 Please submit your revised manuscript by Jul 30 2023 11:59PM. If you will need more time than this to complete your revisions, please reply to this message or contact the journal office at plosone@plos.org. Please include the following items when submitting your revised manuscript:A rebuttal letter that responds to each point raised by the academic editor and reviewer(s). You should upload this letter as a separate file labeled 'Response to Reviewers'.A marked-up copy of your manuscript that highlights changes made to the original version. You should upload this as a separate file labeled 'Revised Manuscript with Track Changes'.An unmarked version of your revised paper without tracked changes. You should upload this as a separate file labeled 'Manuscript'.

We look forward to receiving your revised manuscript.

Kind regards,

Ivone Vaz-Moreira, PhD

Academic Editor

PLOS ONE

“This work was supported by the Korea Ministry of Environment as Projects for Developing Eco-Innovation Technologies (GT-11-G-02-001-3) and by Basic Science Research Program through the National Research Foundation of Korea (NRF) funded by the Ministry of Education (2020R1A6A1A03045059).”

Reviewers' comments:

Reviewer's Responses to Questions

**Comments to the Author**

1. Is the manuscript technically sound, and do the data support the conclusions?

Reviewer #1: Yes

Reviewer #2: Yes

2. Has the statistical analysis been performed appropriately and rigorously? 

Reviewer #1: Yes

Reviewer #2: Yes

3. Have the authors made all data underlying the findings in their manuscript fully available?

Reviewer #1: Yes

Reviewer #2: Yes

4. Is the manuscript presented in an intelligible fashion and written in standard English?

Reviewer #1: Yes

Reviewer #2: Yes

5. Review Comments to the Author

Reviewer #1: 手稿编号 PONE-D-23-10755 ： 主要修订版

这项工作描述了一种基于多粘菌素B（PMB）与GNB外膜脂多糖（LPS）特异性结合的革兰氏阴性菌（GNB）检测方法。GNB最初与表面固定化的PMB结合，然后FITC共轭的PMB与结合的GNB结合，通过发射荧光进行检测。细菌的存在形态包括生物膜细菌和浮游细菌。该方法可以在有限的条件下计数浮游细菌，并且缺乏计数生物膜细菌的实验。此外，我想知道，通过PMB与LPS的特异性结合来计数死细胞是不合理的。由于大多数死菌破裂并将大量LPS分裂到培养基中，在这种情况下，该方法很难通过培养基中PMB与LPS的特异性结合来准确计数含有活菌和死菌的样品。因此，我想知道，作者如何证明该方法可以准确计算临床样本中死菌或活菌的数量。

这些方法依赖于LPS与PMB的结合能力。当使用PBS作为介质时，应评估不同温度和PH对方法准确性的影响。这可以反映方法的稳定性。在我看来，作者的方法无法准确计数细菌死亡和LPS泄漏的标本，生物膜细菌会有很大的误差。作者发现该方法在猪血浆中不准确，但未进行PH或温度等实验。应评价影响多粘菌素B和LPS结合能力的影响因素，这是该方法适用范围的关键点。我建议作者对其进行深度修改，然后以新的面孔重新提交。还有更多评论。

具体评论：

1.第84-87行：菌株的来源和名称不清楚。

2.263-264行：作者说：“所提出的方法既可以检测死GNB，也可以检测活GNB”。我认为这句话缺乏足够的证据。大多数死亡细菌破裂并将大量LPS分裂到培养基中。重要的是要澄清这种方法在这种情况下对死细菌的计数效果。

3.该方法难以通过PMB与培养基中的LPS特异性结合来准确计数含有活菌和死菌的样品。

4.In 图2中，作者没有显示刻度的长度。

5.第294-295行：建议作者探讨猪血浆中成分影响固定化PMB和/或FITC偶联PMB与GNB细胞上GNB细胞结合的机制。

6.PH 通常用于测试结合稳定性。建议作者补充本实验。

Reviewer #2: In this manuscript by Park and colleagues, a microscopy based method for sensitive detection of GNB potentially from patient samples was described. The authors provided good rationale for the experiments and the experiments appear robust and statistically sound.

I only had a couple of follow up questions that would further strengthen the manuscript:

1) Please specify which polymyxin was used - polymyxin B or E?

2) The validation studies utilized an FITC conjugated PMB. Is the conjugation at a specific aa or nonspecific? How variable are the results (RFU detection and sensitivity) using multiple lots of FITC-PMB?

3) The authors should compare different starting densities of GNB vs GPB to ensure that there would not be RFU associated with high concentrations of GPB.

4) The authors should provide a rationale for why porcine plasma was used. What about whole blood as a sample matrix? Can the GNB be detected in whole blood? If not, why not?

6. PLOS authors have the option to publish the peer review history of their article (what does this mean?). If published, this will include your full peer review and any attached files.

Reviewer #1: No

Reviewer #2: No

---

## [Author Response · Author response to Decision Letter 0]

20 Jul 2023

Reviewer 1

Manuscript ID PONE-D-23-10755: Major Revision

This work described a method detecting gram-negative bacteria (GNB) based on specific binding of polymyxin B (PMB) to lipopolysaccharides (LPS) in outer membrane of GNB. GNB were initially bound to surface immobilized PMB, and FITC-conjugated PMB was then bound to the bound GNB for the detection by emiting fluorescence. 

C1. The presence morphology of bacteria includes biofilm bacteria and planktonic bacteria. This method can count bacterioplankton is under limited conditions and lack of experiments for counting biofilm bacteria. 

Response: The authors appreciate for the reviewer’s comment. In this study, we did not investigate sensitivity of the developed method for the biofilms. Instead, we tested the developed method for activated sludge floc consisted of microbial cells and their excreted materials called extra-polymeric substances (EPS), similar to biofilms. The counting sensitivity of the floc was slightly lower than that of the dispersed cells (86% of the dispersed cells), presumably due to difficulty of diffusion of the FITC-conjugated PMB to some microorganisms deeply localized within the floc (See Fig. 5 in the submitted manuscript). In response to the reviewer’s comment, we have not revised manuscript.

Fig. 5. Evaluation of cell aggregation for detecting GNB in the CFU assay and the proposed method. Activated sludge floc was used a form of aggregated cells. The error bars indicate the standard deviation of five measurements.

C2. In addition, I wonder, it is unreasonable to counting dead cells by specific binding of PMB to LPS. Since most of the dead bacteria ruptured and split a large number of LPS into the medium, In that case，this method is difficult to count the samples containing both live bacteria and dead bacteria accurately by specific binding of PMB to LPS in the medium. Therefore, I wonder, how the authors prove that the method could count the number of dead bacteria or live bacteria in the clinical samples accurately. 

Response: The authors thank for the reviewer’s comment. We agree with the reviewer’s speculation that the some dead GNB ruptured and split a large number of LPS into the medium. Nevertheless, we found that the dead GNB possessed enough amount of LPS to be detectable by the PMB method developed in this study. Several studies also reported that some killed GNB still possess LPS (Ringdén, et al., 1977; van Langevelde, et al., 1998). However, the counting sensitivity of the dead cells represented by fluorescence intensity was slightly lower than that of live cells (91% of the live cells), presumably due to the reason speculated by the reviewer. In response to the reviewer’s comment, we have addressed the lower sensitivity for the dead cells, as follows. 

Page 13, line 271-273:

“The counting sensitivity of the dead cells represented by fluorescence intensity was slightly lower than that of live cells (91% of the live cells), presumably due to some dead E. coli cells ruptured and split LPS into the medium.”

Fig. 6. Evaluation of viability of for detecting GNB in the CFU assay and the proposed method. E. coli cells were killed using 0.5% sodium hypochlorite for a preparation of dead GNB cells. The error bars indicate the standard deviation of five measurements.

Reference

Ringdén, O., Rynnel‐Dagöö, B., Waterfield, E. M., Möller, E., & Möller, G. (1977). Polyclonal antibody secretion in human lymphocytes induced by killed staphylococcal bacteria and by lipopolysaccharide. Scandinavian Journal of Immunology, 6(11), 1159-1169.

van Langevelde, P., Kwappenberg, K. M., Groeneveld, P. H., Mattie, H., & van Dissel, J. T. (1998). Antibiotic-induced lipopolysaccharide (LPS) release from Salmonella typhi: delay between killing by ceftazidime and imipenem and release of LPS. Antimicrobial agents and chemotherapy, 42(4), 739-743.

C3. The methods relied on the binding ability of LPS to PMB. When using PBS as a medium, the effect of different temperatures and PH on the accuracy of the method should be evaluated. This can reflect the stability of the method. In my opinion, the authors' method cannot accurately count specimens with bacterial death and LPS leakage and there will be a large error in biofilm bacteria. The authors found that the method was inaccurate in porcine plasma，but did not perform experiment such as PH or temperature. The influencing factors affecting the binding ability of polymyxin B and LPS should be evaluated which are the key points to the scope of application of this method. I recommend that authors deeply revise it and then resubmit it in a new face. There are more comments.

Response: The authors appreciate for the reviewer’s important comment and agree with the opinion. During optimization of the PMB-based method, we already tested the binding sensitivity between PMB and LPS under different pH and temperature conditions to optimize this newly developed method, but we did not include the results in the submitted manuscript. As shown in the figure below, sensitivities of the PMB method were not significantly affected by different pH and temperature for various GNB. In response to the reviewer’s comment, we have addressed the effects of temperature and pH in the revised manuscript. In addition, we have added the figures in the supplementary materials.

Page 11, line 228-231:

“pH and temperature may affect the sensitivity of the proposed PMB method, but experiments conducted for different pH (3, 7, 11) and temperature (4, 36, 60oC) for various GNB demonstrated insensitivity of the PMB method (Supplementary Fig. 1).”

Supplementary Fig. 1. Sensitivity of the proposed PMB method under different (A) pH and (B) temperature conditions. The error bars indicate the standard deviation of five measurements.

Specific comments:

Q1. Line 84-87: The source and name of the strains are not clear.

Response: The authors thank for the reviewer’s careful comment. In response to the reviewer’s comment, we have added the source and name of bacterial strains used in this study in the revised manuscript.

Page 5, line 84-88:

“Five species of GNB (Escherichia coli strain O111:B4, Pseudomonas aeruginosa PA14, Sphingomonas paucimobilis ATCC51231, Pseudomonas alcaliphila AL15-21T (JCM 10630T), and Enterobacter cloacae ATCC13047) and four species of GPB (Staphylococcus aureus ATCC6538, Bacillus amyloliquefaciens strain FXH73, Bacillus thuringiensis ATCC10792, and Corynebacterium xerosis ATCC7711) were used in this study.”

Fig. 2. Verification of the proposed method detecting GNB. a) CLSM images of DAPI (left), FITC (middle), and overlay of the two images (right) for the E. coli cells processed by the proposed method and stained with DAPI. Scale bars indicate 30 μm. b) FITC signal intensity for various GPB and GNB processed by the proposed method. Control indicates a sample without bacterial cells. The error bars indicate the standard deviation of five measurements. *, P > 0.10 versus control. 

Q2. Line 263-264: The author said: “the proposed method can detect dead GNB as well as live GNB”. I think the sentence lack of sufficient evidence. Most of the dead bacteria ruptured and split a large number of LPS into the medium. It is important to clarify the counting effect of this method on dead bacteria in such condition.

Response: The authors thank for the reviewer’s comment. We agree with the reviewer’s speculation that the some dead GNB ruptured and split a large number of LPS into the medium. Nevertheless, we found that the dead GNB possessed enough amount of LPS to be detectable by the PMB method developed in this study. Several studies also reported that some killed GNB still possess LPS (Ringdén, et al., 1977; van Langevelde, et al., 1998). However, the counting sensitivity of the dead cells represented by fluorescence intensity was slightly lower than that of live cells (91% of the live cells), presumably due to the reason speculated by the reviewer. In response to the reviewer’s comment, we have addressed the lower sensitivity for the dead cells, as follows. 

Page 13, line 271-273

“The counting sensitivity of the dead cells represented by fluorescence intensity was slightly lower than that of live cells (91% of the live cells), presumably due to some dead E. coli cells ruptured and split LPS into the medium.”

Fig. 6. Evaluation of viability of for detecting GNB in the CFU assay and the proposed method. E. coli cells were killed using 0.5% sodium hypochlorite for a preparation of dead GNB cells. The error bars indicate the standard deviation of five measurements.

Reference

Ringdén, O., Rynnel‐Dagöö, B., Waterfield, E. M., Möller, E., & Möller, G. (1977). Polyclonal antibody secretion in human lymphocytes induced by killed staphylococcal bacteria and by lipopolysaccharide. Scandinavian Journal of Immunology, 6(11), 1159-1169.

van Langevelde, P., Kwappenberg, K. M., Groeneveld, P. H., Mattie, H., & van Dissel, J. T. (1998). Antibiotic-induced lipopolysaccharide (LPS) release from Salmonella typhi: delay between killing by ceftazidime and imipenem and release of LPS. Antimicrobial agents and chemotherapy, 42(4), 739-743.

Q3. This method is difficult to count the samples containing both live bacteria and dead bacteria accurately by specific binding of PMB to LPS in the medium.

Response: The authors thank for the reviewer’s comment. We agree with the reviewer’s opinion that the PMB method cannot distinguish between live and dead GNB. As addressed in the submitted manuscript, we speculate that the feature can be a merit for testing clinical samples. LPS are major endotoxins that can elicit various immune responses via binding with toll-like receptors in human [31]. As such immune responses are initiated by LPS that are released by mostly dead GNB [8, 31], dead rather than live GNB pose more of a threat to human beings; thus, proper attention must be paid to dead GNB as well. In this respect, capability of dead GNB detection by the proposed method suggests it can be used for early detection of such risk elements. In response to this comment, we have not revised the manuscript.

Reference

[8] DeLucca AJ, Godshall MA, Palmgren MS. Gram-negative bacterial endotoxins in grain elevator dusts. American Industrial Hygiene Association Journal. 1984;45(5):336-9.

[31] Heumann D, Roger T. Initial responses to endotoxins and Gram-negative bacteria. Clinica Chimica Acta. 2002;323(1-2):59-72. 

Q4. In Fig2, the authors did not show the length of the scale.

Response: The authors thank for the reviewer’s comment. In response to the reviewer’s comment, we have added the length of scale in the revised manuscript.

Fig. 2. Verification of the proposed method detecting GNB. a) CLSM images of DAPI (left), FITC (middle), and overlay of the two images (right) for the E. coli cells processed by the proposed method and stained with DAPI. Scale bars indicate 30 μm. b) FITC signal intensity for various GPB and GNB processed by the proposed method. Control indicates a sample without bacterial cells. The error bars indicate the standard deviation of five measurements. *, P > 0.10 versus control. 

Q5. Line 294-295: It is suggested that the authors explore the mechanism of the components in porcine plasma affected the binding of GNB cells on the immobilized PMB and/or FITC conjugated PMB to GNB cells.

Response: The authors appreciate the reviewer’s question. Identification of the mechanism to lower the linearity in the test using porcine plasma compared with the test using PBS is beyond the scope of this research. However, we speculate that the component of porcine plasma such as proteins, nutrients, waste products, and electrolytes might affect the binding between PMB and LPS interaction. In response to the reviewer’s comment, we have added a sentence in the revised manuscript, as follows.

Page 14, line 300-301:

“Porcine plasma is reported to contain various proteins, nutrients, and electrolytes [32].”

Reference

[32] Engelking, L. R. (2015). Chapter 4-Protein Structure. Textbook of Veterinary Physiological Chemistry, 3, 18-25.

Q6. pH is commonly used to test the binding stability. The authors are advised to supplement this experiment.

Response: The authors agree with the reviewer’s comment on the importance of pH in the binding stability. As mentioned earlier, we have added the experimental result on different pH condition, as follow.

Page 11, line 228-231:

“pH and temperature may affect the sensitivity of the proposed PMB method, but experiments conducted for different pH (3, 7, 11) and temperature (4, 36, 60oC) for various GNB demonstrated insensitivity of the PMB method (Supplementary Fig. 1).”

Supplementary Fig. 1. Sensitivity of the proposed PMB method under different (A) pH and (B) temperature conditions. The error bars indicate the standard deviation of five measurements. 

Reviewer 2

In this manuscript by Park and colleagues, a microscopy-based method for sensitive detection of GNB potentially from patient samples was described. The authors provided good rationale for the experiments and the experiments appear robust and statistically sound.

I only had a couple of follow up questions that would further strengthen the manuscript:

Q1. Please specify which polymyxin was used - polymyxin B or E?

Response: The authors thank for the reviewer’s kind comment. We used polymyxin B in the whole experiment. In response to the reviewer's comment, we have specified the polymyxin as “polymyxin B” in the revised manuscript.

Q2. The validation studies utilized an FITC conjugated PMB. Is the conjugation at a specific aa or nonspecific? How variable are the results (RFU detection and sensitivity) using multiple lots of FITC-PMB?

Response: The authors thank for the reviewer’s comment. We have used FITC using the LYNX rapid fluorescein antibody conjugation kit (Bio-Rad, Hercules, CA, USA) for generating FITC-PMB conjugation. When the kit used, FITC was binding to the primary amines and thiols (e.g. thiomersal/thimerosal) of PMB during the procedures. We addressed the conjugation method in the Materials and Method section. Additionally, the variability of FITC conjugation efficiency was not observed using multiple lots of FITC-PMB in our experiments. 

Q3. The authors should compare different starting densities of GNB vs GPB to ensure that there would not be RFU associated with high concentrations of GPB.

Response: The authors appreciate for the reviewer’s critical comment. Unfortunately, we did not conduct experiments for different ratio between GNB and GPB. One reason of not to conduct such experiments was that several researchers reported that PMB have no affinity with GPB possessing a thick peptidoglycan layer and the absence of the outer membrane (Yin et al., 2020; Evans et al., 1999). Another reason was that the signal intensities were very low and not significantly different from those of the control (i.e., no cell condition), even under very high cell concentration (107 cells/mL) for the various GNB (see below Fig. 2 (B)). 

Reference

Yin, J., Meng, Q., Cheng, D., Fu, J., Luo, Q., Liu, Y., & Yu, Z. (2020). Mechanisms of bactericidal action and resistance of polymyxins for Gram-positive bacteria. Applied microbiology and biotechnology, 104, 3771-3780.

Evans, M. E., Feola, D. J., & Rapp, R. P. (1999). Polymyxin B sulfate and colistin: old antibiotics for emerging multiresistant gram-negative bacteria. Annals of Pharmacotherapy, 33(9), 960-967.

Fig. 2. FITC signal intensity for various GPB and GNB processed by the proposed method. Control indicates a sample without bacterial cells. The error bars indicate the standard deviation of five measurements. *, P > 0.10 versus control. 

Q4. The authors should provide a rationale for why porcine plasma was used. What about whole blood as a sample matrix? Can the GNB be detected in whole blood? If not, why not?

Response: The authors appreciate for the reviewer’s comment. Although we have not included the result in the submitted manuscript, we conducted detection sensitivity of the PMB method using whole blood. As shown a figure below, the result showed that sensitivity as well as linearity of the relationship between cell numbers and fluorescent intensity were lower than those with porcine plasma.

---

## [Decision Letter · Decision Letter 1]

11 Aug 2023

Rapid and sensitive detection of gram-negative bacteria using surface-immobilized polymyxin B

PONE-D-23-10755R1

Dear Dr. Park,

We’re pleased to inform you that your manuscript has been judged scientifically suitable for publication and will be formally accepted for publication once it meets all outstanding technical requirements.

Kind regards,

Ivone Vaz-Moreira, PhD

Academic Editor

PLOS ONE

Additional Editor Comments (optional):

Reviewers' comments:

Reviewer's Responses to Questions

**Comments to the Author**

1. If the authors have adequately addressed your comments raised in a previous round of review and you feel that this manuscript is now acceptable for publication, you may indicate that here to bypass the “Comments to the Author” section, enter your conflict of interest statement in the “Confidential to Editor” section, and submit your "Accept" recommendation.

Reviewer #1: All comments have been addressed

2. Is the manuscript technically sound, and do the data support the conclusions?

Reviewer #1: Yes

3. Has the statistical analysis been performed appropriately and rigorously? 

Reviewer #1: Yes

4. Have the authors made all data underlying the findings in their manuscript fully available?

Reviewer #1: Yes

5. Is the manuscript presented in an intelligible fashion and written in standard English?

Reviewer #1: Yes

6. Review Comments to the Author

Reviewer #1: (No Response)

7. PLOS authors have the option to publish the peer review history of their article (what does this mean?). If published, this will include your full peer review and any attached files.

Reviewer #1: No

---

## [Editor Report · Acceptance letter]

18 Aug 2023

PONE-D-23-10755R1 

Rapid and sensitive detection of gram-negative bacteria using surface-immobilized polymyxin B 

Dear Dr. Park:

I'm pleased to inform you that your manuscript has been deemed suitable for publication in PLOS ONE. Congratulations! Your manuscript is now with our production department. 

Kind regards, 

on behalf of

Dr. Ivone Vaz-Moreira 

Academic Editor

PLOS ONE